# Involvement of the γ Isoform of cPLA_2_ in the Biosynthesis of Bioactive *N-*Acylethanolamines

**DOI:** 10.3390/molecules26175213

**Published:** 2021-08-27

**Authors:** Yiman Guo, Toru Uyama, S. M. Khaledur Rahman, Mohammad Mamun Sikder, Zahir Hussain, Kazuhito Tsuboi, Minoru Miyake, Natsuo Ueda

**Affiliations:** 1Department of Biochemistry, Kagawa University School of Medicine, 1750-1 Ikenobe, Miki, Kagawa 761-0793, Japan; s18d710@stu.kagawa-u.ac.jp (Y.G.); uyama.toru@kagawa-u.ac.jp (T.U.); smk.rahman@just.edu.bd (S.M.K.R.); s20d710@stu.kagawa-u.ac.jp (M.M.S.); MDH106@pitt.edu (Z.H.); 2Department of Oral and Maxillofacial Surgery, Kagawa University School of Medicine, 1750-1 Ikenobe, Miki, Kagawa 761-0793, Japan; dentmm@med.kagawa-u.ac.jp; 3Department of Pharmacology, Kawasaki Medical School, 577 Matsushima, Kurashiki, Okayama 701-0192, Japan; ktsuboi@med.kawasaki-m.ac.jp

**Keywords:** *N-*acyltransferase, anandamide, endocannabinoid, phospholipase A_2_

## Abstract

Arachidonylethanolamide (anandamide) acts as an endogenous ligand of cannabinoid receptors, while other *N-*acylethanolamines (NAEs), such as palmitylethanolamide and oleylethanolamide, show analgesic, anti-inflammatory, and appetite-suppressing effects through other receptors. In mammalian tissues, NAEs, including anandamide, are produced from glycerophospholipid via *N-*acyl-phosphatidylethanolamine (NAPE). The ɛ isoform of cytosolic phospholipase A_2_ (cPLA_2_) functions as an *N-*acyltransferase to form NAPE. Since the cPLA_2_ family consists of six isoforms (α, β, γ, δ, ɛ, and ζ), the present study investigated a possible involvement of isoforms other than ɛ in the NAE biosynthesis. Firstly, when the cells overexpressing one of the cPLA_2_ isoforms were labeled with [^14^C]ethanolamine, the increase in the production of [^14^C]NAPE was observed only with the ɛ-expressing cells. Secondly, when the cells co-expressing ɛ and one of the other isoforms were analyzed, the increase in [^14^C]*N*-acyl-lysophosphatidylethanolamine (lysoNAPE) and [^14^C]NAE was seen with the combination of ɛ and γ isoforms. Furthermore, the purified cPLA_2_γ hydrolyzed not only NAPE to lysoNAPE, but also lysoNAPE to glycerophospho-*N*-acylethanolamine (GP-NAE). Thus, the produced GP-NAE was further hydrolyzed to NAE by glycerophosphodiesterase 1. These results suggested that cPLA_2_γ is involved in the biosynthesis of NAE by its phospholipase A_1_/A_2_ and lysophospholipase activities.

## 1. Introduction

*N*-Acylethanolamines (NAEs) are a class of bioactive lipids consisting of long-chain fatty acids and ethanolamine, and are widely present in animal and plant tissues [1]. They exhibit different biological activities depending on the type of constituent fatty acids. For example, *N*-arachidonoylethanolamine, also called arachidonylethanolamide or anandamide, functions as an endocannabinoid that binds to cannabinoid receptors CB1 and CB2 [2]. On the other hand, *N*-palmitoylethanolamine (palmitylethanolamide) [3] and *N*-oleoylethanolamine (oleylethanolamide) [4] act on the peroxisome proliferator-activated receptor (PPAR)-α as well as other receptors to show anti-inflammatory/analgesic and appetite-suppressing effects, respectively.

NAEs are biosynthesized from membrane phospholipids mainly in two-step enzyme reactions (Figure 1) [5,6]. The first reaction is the transfer of a fatty acyl chain from the *sn*-1 position of a glycerophospholipid molecule such as phosphatidylcholine (PC) to the amino group of a diacyl-type or plasmalogen-type phosphatidylethanolamine (PE), resulting in the formation of *N*-acyl-phosphatidylethanolamine (NAPE), a unique phospholipid molecule with three fatty acyl chains. The enzymes catalyzing this reaction are collectively called *N*-acyltransferases, which are classified into two groups by Ca^2+^-dependency [5,6,7,8]. The second reaction is the release of NAE from NAPE. The phospholipase D (PLD)-type enzyme NAPE-PLD directly produces NAE [9], while the alternative pathway does not involve NAPE-PLD, but consists of consecutive hydrolytic reactions via *N*-acyl-lysophosphatidylethanolamine (lysoNAPE) and glycerophospho-*N*-acylethanolamine (GP-NAE) (Figure 1) [10,11]. This pathway involves several hydrolases such as group IB, IIA, and V of secretory phospholipase A_2_s (sPLA_2_s) [12], α/β-hydrolase domain containing 4 (ABHD4) [13], and glycerophosphodiesterase (GDE) 1 [14]. The analysis of NAPE-PLD-deficient mice demonstrated the presence of the alternative pathway in the brain [11,15] and peripheral tissues such as heart, kidney, liver, and jejunum [16].

The cytosolic phospholipase A_2_ (cPLA_2_) family, also referred to as the group IV PLA_2_ family, belongs to the PLA_2_ superfamily and consists of six isoforms (α, β, γ, δ, ɛ, and ζ) [17]. Ogura et al. revealed that the ε isoform of the cPLA_2_ family (cPLA_2_ε), also known as group IVE PLA_2_ (PLA2G4E), functions as a Ca^2+^-dependent *N*-acyltransferase to form NAPE [8]. On the other hand, the involvement of isoforms other than ε in the biosynthetic pathway of NAE has not yet been reported. In the present study, we examined the facilitatory effects of the isoforms of the cPLA_2_ family on the NAE formation in living mammalian cells, as well as the reactivity of purified cPLA_2_γ with NAPE and lysoNAPE. The results suggested that the γ isoform deacylates NAPE to GP-NAE via the formation of lysoNAPE and constitutes the alternative pathway for the formation of NAE.

## 2. Results and Discussion

### 2.1. N-Acyltransferase Activity of cPLA_2_ Isoforms in Living Cells

To examine whether mouse cPLA_2_ isoforms have NAPE-producing *N*-acyltransferase activity, we transiently expressed each isoform (α, β, γ, δ, ɛ, or ζ) in human embryonic kidney (HEK)293 cells. Since these recombinant proteins were tagged with FLAG, their successful expression was confirmed by Western blotting using anti-FLAG antibody (Figure 2A). These FLAG-tagged proteins exhibited immunopositive bands at the position of the deduced molecular mass of each isoform (α, 85 kDa; β, 88 kDa; γ, 68 kDa; δ, 93 kDa; ɛ, 100 kDa; and ζ, 96 kDa), respectively. The extra bands were presumed to be degradative or modified proteins of each isoform.

We recently reported that cPLA_2_ɛ-expressing cells pretreated with [^14^C]ethanolamine produce a large amount of [^14^C]NAPE in response to the Ca^2+^ ionophore ionomycin [18]. Thus, in the present study, we used this metabolic labeling as a simple method to detect Ca^2+^-dependent *N*-acyltransferase activity in living cells. We cultured the HEK293 cells expressing each isoform in the presence of [^14^C]ethanolamine for 18 h and further treated the cells with ionomycin for 30 min. Total lipids were then extracted from the cells and separated by thin-layer chromatography (TLC). The distribution of radioactivity on the thin-layer plate was visualized (Figure 2B) and quantified (Figure 2C–E). In comparison with the control cells transfected with an insert-free vector, the cPLA_2_ɛ-expressing cells exhibited remarkable increases in the intensities of the radioactive bands corresponding to authentic NAPE, NAE, and lysoNAPE. However, such increases were not seen with the cells expressing cPLA_2_ isoforms other than ɛ.

The δ isoform (PLA2G4D) was previously reported not to exhibit *N*-acyltransferase activity [8]. Although our results suggested that the ε isoform is the sole enzyme functioning in living cells as NAPE-forming *N*-acyltransferase, we could not rule out the possibility that the other isoforms show *N*-acyltransferase activity under different assay conditions.

### 2.2. NAPE-PLA_1_/A_2_ Activity of cPLA_2_ Isoforms in Living Cells

Since all the cPLA_2_ isoforms were previously reported to exhibit PLA_1_/A_2_ activity for glycerophospholipids such as PC [17], it was likely that these isoforms also showed PLA_1_/A_2_ activity for NAPE in living cells. For this purpose, we used cPLA_2_ɛ/Tet-on cells, which stably expressed FLAG-tagged cPLA_2_ɛ in the presence of doxycycline (DOX) and produced a large amount of NAPE when stimulated by a Ca^2+^ ionophore [18]. We transiently expressed one of α, β, γ, δ, and ζ isoforms with a FLAG tag in cPLA_2_ɛ/Tet-on cells in the presence of DOX. Successful co-expression of ɛ isoform and one of the other isoforms was confirmed by Western blotting using anti-FLAG antibody (Figure 3A). cPLA_2_ε was stably expressed in cPLA_2_ε/Tet-on cells in the presence of doxycyclin, while other isoforms of cPLA_2_ were transiently and potently expressed by the introduction of each cDNA using Lipofectamine 2000. The differences in the expression levels between ε and other isoforms were presumably attributed to the differences in the expression methods. Metabolic labeling of cPLA_2_ɛ/Tet-on cells with [^14^C]ethanolamine, followed by the ionomycin treatment, exhibited the production of large amounts of radioactive NAPE, NAE, and lysoNAPE due to the activation of cPLA_2_ɛ (Figure 3B) as reported previously [18]. Interestingly, as compared with the sole expression of ɛ, the co-expression of ɛ with γ, but not with α, β, δ, or ζ, showed a lower level of NAPE (Figure 3C) and higher levels of NAE (Figure 3D) and lysoNAPE (Figure 3E). These results suggested that NAPE, produced by ɛ, was hydrolyzed to lysoNAPE by the PLA_1_/A_2_ activity of γ. The increase in NAE levels by the expression of γ was presumably due to further hydrolysis of the increased lysoNAPE.

Earlier, mouse ABHD4 was reported to have the ability to hydrolyze NAPE to lysoNAPE, and then lysoNAPE to GP-NAE (Figure 1) [13]. In fact, substantial reductions in GP-NAE and plasmalogen-type lysoNAPE were observed in the brain of ABHD4-deficient mice [19]. Thus, we also transiently expressed mouse ABHD4 in cPLA_2_ɛ/Tet-on cells. Western blotting revealed the expression of FLAG-tagged ABHD4 with a molecular mass of 39 kDa (Figure 3A). However, in the metabolic labeling with [^14^C]ethanolamine, the expression of ABHD4 did not significantly affect the levels of radioactive NAPE, NAE, or lysoNAPE (Figure 3B–E), despite the fact that the purified ABHD4 successfully hydrolyzed *N-*[^14^C]palmitoyl-PE to *N-*[^14^C]palmitoyl-lysoPE (Figure 4A,B). The reason for this discrepancy remained unclear.

### 2.3. Activities of Purified cPLA_2_γ and ABHD4

We expressed FLAG-tagged cPLA_2_γ and ABHD4 in HEK293 cells and purified these enzymes by anti-FLAG affinity chromatography. The purified enzymes were allowed to react with *N-*[^14^C]palmitoyl-PE, and the radioactive products were separated by TLC (Figure 4A). The results showed that both enzymes produced two radioactive bands corresponding to *N-*[^14^C]palmitoyl-lysoPE (Figure 4B) and GP-*N-*[^14^C]palmitoylethanolamine (Figure 4C). Moreover, when the purified enzymes were incubated with *N-*[^14^C]palmitoyl-lysoPE, the production of GP-*N-*[^14^C]palmitoylethanolamine was observed (Figure 5A,B). These results showed that the purified cPLA_2_γ, as well as the purified ABHD4 catalyzed two sequential hydrolytic reactions to convert NAPE to GP-NAE via lysoNAPE. Notably, cPLA_2_γ catalyzed the latter reaction at a higher rate than the former reaction, suggesting the efficient formation of GP-NAE from NAPE.

cPLA_2_γ was earlier cloned from a human [20] and characterized as a novel membrane-bound, Ca^2+^-independent PLA_2_ [17,20]. Our preliminary assay also showed that the purified cPLA_2_γ can hydrolyze [^14^C]PC (data not shown). In addition to PLA_2_ activity, cPLA_2_γ exhibited PLA_1_, lysophospholipase, and acyltransferase activity [21]. The substrates used were PC, PE, lysoPC, and lysoPE. Thus, cPLA_2_γ sequentially hydrolyzed two acyl chains from *sn*-1 and -2 positions of the glycerol backbone of PC and PE, resulting in the formation of glycerophosphocholine or glycerophosphoethanolamine, respectively. The ability to convert NAPE to GP-NAE via lysoNAPE (Figure 4 and Figure 5) may be explained by this multi-function of cPLA_2_γ.

To identify one of the cPLA_2_γ products as GP-*N-*[^14^C]palmitoylethanolamine, we extracted this radioactive product from silica gel with organic solvent and allowed the substance to react with purified recombinant mouse GDE1, which is known to hydrolyze GP-NAE to NAE and glycerol 3-phoshate [14]. As shown in Figure 6A, GDE1 converted the substance to a radioactive band corresponding to authentic *N-*[^14^C]palmitoylethanolamine. Furthermore, we incubated the purified cPLA_2_γ with *N-*[^14^C]palmitoyl-PE for 30 min and then added the purified GDE1 to the reaction mix, followed by further incubation for 15 min (Figure 6B–E). This sequential reaction led to the production of the radioactive band corresponding to *N-*[^14^C]palmitoylethanolamine. In contrast, GDE1 was inactive with *N-*[^14^C]palmitoyl-PE. These results suggest that GP-NAE produced by cPLA_2_γ is converted to NAE by GDE1.

### 2.4. Tissue Distributions of cPLA_2_γ and ABHD4

We examined the distribution of mRNAs of cPLA_2_γ and ABHD4 in mouse tissues by reverse transcription-PCR (Figure 7A). cPLA_2_γ mRNA was widely distributed in various tissues with higher expression levels in the liver, colon, and testis, followed by many other tissues (Figure 7B). On the other hand, ABHD4 mRNA was widely distributed with higher levels in the brain, heart, lung, ileum, kidney, testis, and skeletal muscle (Figure 7C). Previously, the tissue distribution of mouse ABHD4 mRNA was reported with the highest expression in the central nervous system and testis, followed by the liver and kidney, with negligible signals in the heart [13].

In the nervous system, ABHD4 did not appear to be the sole enzyme that hydrolyzed NAPE and lysoNAPE [19]. Moreover, the enzyme(s) responsible in peripheral tissues have not fully been understood. Considering the wide distribution of cPLA_2_γ in mouse tissues, cPLA_2_γ may function as an alternative of ABHD4 in the NAE biosynthesis. Specific inhibitors for these enzymes will be useful to quantitatively estimate the contribution of each enzyme, particularly if used in primary culture.

As for human cPLA_2_γ, Northern blot analysis indicated that cPLA_2_γ mRNA is most abundant in the skeletal muscle and heart, with lower levels in the spleen, brain, placenta, and pancreas [20]. Human cPLA_2_γ (accession number, NP_003697) [20] and mouse cPLA_2_γ (NM_001004762) were deduced to comprise 541 and 597 amino acids, respectively. Arg-54, Ser-82, and Asp-385, forming the catalytic center of human cPLA_2_γ, were conserved as Arg-55, Ser-83, and Asp-417 in mouse cPLA_2_γ [17]. Human cPLA_2_γ [20] and mouse cPLA_2_γ (Guo et al., unpublished observation) showed the PLA_1_/A_2_ activity for PC. Thus, human cPLA_2_γ was considered to be the ortholog of the mouse enzyme.

## 3. Materials and Methods

### 3.1. Materials

[1,2-^14^C]Ethanolamine-HCl ([^14^C]ethanolamine) was purchased from Moravek Biochemicals (Brea, CA, USA); anti-FLAG M2-conjugated agarose affinity gel and FLAG peptide were from Sigma-Aldrich (St. Louis, MO, USA); rabbit anti-FLAG (DYKDDDDK) monoclonal antibody was from Cell Signaling Technology (Danvers, MA, USA); horseradish peroxidase-linked anti-rabbit IgG was from GE Healthcare (Piscataway, NJ, USA); protein assay dye reagent concentrate was from Bio-Rad (Hercules, CA, USA); PrimeScript RT reagent kit was from Takara Bio (Kusatsu, Japan); precoated silica gel 60 F254 aluminum sheets for TLC (20 × 20 cm, 0.2 mm thickness) and Immobilon-P were from Merck Millipore (Darmstadt, Germany); fetal bovine serum, Lipofectamine 2000, TRIzol, pEF6/Myc-His vector, and Pierce Western Blotting Substrate Plus were from Invitrogen/Thermo Fisher Scientific (Carlsbad, CA, USA); Nonidet P-40 was from Nacalai Tesque (Kyoto, Japan); Dulbecco’s modified Eagle’s medium (DMEM), dithiothreitol (DTT), 3(2)-*t*-butyl-4-hydroxyanisole (BHA), Tween 20, and ionomycin were from FUJIFILM Wako Pure Chemical (Osaka, Japan); KOD-Plus-Neo polymerase and Quick taq DNA polymerase were from TOYOBO (Osaka, Japan); *n*-octyl-β-D-glucoside and 3-[(3-cholamidopropyl)dimethylammonio]-propanesulfonate (CHAPS) were from Dojindo (Kumamoto, Japan); DOX was from Clontech (Mountain View, CA, USA); HEK293 cells were from Health Science Research Resources Bank (Osaka, Japan). 1,2-Dioleoyl-*sn-*glycero-3-phospho(*N-*[1′-^14^C]palmitoyl)ethanolamine (*N-*[^14^C]palmitoyl-PE), 1-oleoyl-2-hydroxy-*sn-*glycero-3-phospho(*N-*[1′-^14^C]palmitoyl)ethanolamine (*N-*[^14^C]palmitoyl-lysoPE), *sn-*glycero-3-phospho(*N-*[1′-^14^C]palmitoyl)ethanolamine (glycerophospho-*N-*[^14^C]palmitoylethanolamine or GP-*N-*[^14^C]palmitoylethanolamine) and *N-*[^14^C]palmitoylethanolamine were enzymatically prepared as described previously [22]. The products were purified by TLC with a mixture of chloroform/methanol/28% ammonium hydroxide (80:20:2, by vol.) or chloroform/methanol/acetic acid (9:1:1, by vol.).

### 3.2. Construction of Expression Vectors

C57BL/6 mice (male, 8 weeks old) (Japan SLC, Inc., Hamamatsu, Japan) were anesthetized and sacrificed by decapitation according to the guidelines for care and use of animals established by Kagawa University (Kagawa, Japan). Total RNAs were then isolated using TRIzol from the mouse tissues indicated in Table 1. First-strand cDNA was prepared from 5 μg of total RNA using a PrimeScript RT reagent kit. The cDNA encoding N-terminally FLAG-tagged mouse cPLA_2_α, cPLA_2_β, cPLA_2_γ, cPLA_2_δ, cPLA_2_ζ, and ABHD4 was amplified by PCR with KOD-Plus-Neo DNA polymerase. The primers used are shown in Table 1. PCR was carried out for 35 cycles at 94 °C for 20 s, 56 °C for 20 s, and 68 °C for 3 min. The obtained DNA fragments were subcloned into the corresponding sites of pEF6/Myc-His. All the constructs were sequenced in both directions using an ABI 3130 Genetic Analyzer (Invitrogen/Life Technologies, Carlsbad, CA, USA). The expression vectors harboring N-terminally FLAG-tagged mouse cPLA_2_ε [23] and C-terminally FLAG-tagged mouse GDE1 [22] were constructed as described previously.

### 3.3. Metabolic Labeling

A Tet-on cell line (FL-cPLA_2_ε/Tet-on), which DOX-dependently expresses FLAG-tagged cPLA_2_ε, was established by the transfection of HEK293 cells with pcDNA5/TO vector harboring FLAG-tagged cPLA_2_ε as reported previously [18]. The cells were maintained for at least four days in the presence of 1 μg/mL DOX.

HEK293 cells and FL-cPLA_2_ε/Tet-on cells were grown at 37 °C to 80% confluency in 6-well plastic plates containing DMEM with 10% fetal bovine serum in a humidified 5% CO_2_ and 95% air incubator. For the transient expression of FLAG-tagged enzymes, the expression vectors harboring cDNA of each enzyme were introduced into the cells using Lipofectamine 2000 according to the manufacturer’s instructions. Twenty-four hours after transfection, the cells were labeled with [^14^C]ethanolamine (0.16 μCi/well) for 18 h. [^14^C]Ethanolamine was then removed and serum-free fresh medium with or without 2 μM ionomycin was added to the wells. After further incubation at 37 °C for 30 min, total lipids were extracted by the method of Bligh and Dyer [24], spotted on a silica gel thin-layer plate (20 cm height), and developed at 4 °C for 90 min with a mixture of chloroform/methanol/28% ammonium hydroxide (80:20:2, by vol.). The distribution of radioactivity on the plate was visualized and quantified using an image reader FLA-7000 (FUJIFILM, Tokyo, Japan). All assays were performed in triplicate.

### 3.4. Expression and Purification of Recombinant Proteins

HEK293 cells were grown at 37 °C to 90% confluency in 150 mm plastic dishes containing DMEM with 10% fetal bovine serum in a humidified 5% CO_2_ and 95% air incubator. For the expression of recombinant FLAG-tagged cPLA_2_γ, ABHD4, or GDE1, their expression vectors were introduced into HEK293 cells using Lipofectamine 2000 according to the manufacturer’s instructions. Forty-eight hours after transfection, cells were harvested from 2 to 3 dishes and sonicated twice each for 5 s in 20 mM Tris-HCl (pH 7.4).

For the purification of cPLA_2_γ and ABHD4, soluble fractions were prepared from the cell homogenates by centrifugation in the presence of 0.1% Nonidet P-40 at 105,000× *g* for 30 min at 4 °C; they were then mixed with 1 mL of a 50% slurry of anti-FLAG M2 affinity gel pre-equilibrated with 50 mM Tris-HCl (pH 7.4) containing 150 mM NaCl and 0.05% Nonidet P-40 (buffer A). After overnight incubation at 4 °C under gentle mixing, the gel was packed into a column and washed three times each with 12 mL of buffer A. The FLAG-tagged protein was eluted with buffer A containing 0.1 mg/mL of FLAG peptide, and every 0.25 mL fraction was collected.

For the purification of GDE1, particulate fractions were prepared from the homogenates of the GDE1-expressing cells by centrifugation in the presence of 0.1% octyl glucoside at 105,000× *g* for 30 min at 4 °C. GDE1 was then solubilized and purified as described previously [22].

The protein concentration was determined by the method of Bradford with bovine serum albumin as a standard.

### 3.5. Enzyme Assay

Purified cPLA_2_γ (2 µg protein) and ABHD4 (2 µg protein) were allowed to react with 25 μM *N-*[^14^C]palmitoyl-PE (25,000 cpm, dissolved in 5 μL ethanol) or 25 μM *N-*[^14^C]palmitoyl-lysoPE (25,000 cpm, dissolved in 5 μL ethanol) in 100 μL of 50 mM Tris-HCl (pH 7.4), 0.1% CHAPS, and 5 mM EDTA at 37 °C for 30 min. The reaction was terminated by adding 0.32 mL of chloroform/methanol/1 M citric acid (8:4:1, *v*/*v*) containing 5 mM BHA. After centrifugation, 100 μL of the organic phase was spotted on a thin-layer silica gel plate (20 cm height), and developed in chloroform/methanol/water (65:25:4, by vol.) at 4 °C for 90 min.

Purified GDE1 (2 µg protein) was incubated with ^14^C-labeled compounds in 100 μL of 50 mM Tris-HCl (pH 7.4), 3 mM DTT, and 2 mM MgCl_2_ at 37 °C for 30 min. The reaction was terminated by adding 0.32 mL of chloroform/methanol/1 M citric acid (8:4:1, by vol.) containing 5 mM BHA. After centrifugation, 100 μL of the organic phase was spotted on a thin-layer silica gel plate (20 cm height), and developed in chloroform/methanol/28% ammonium hydroxide (80:20:2, by vol.) at 4 °C for 90 min.

After the development by TLC, the radioactive substances on the plate were quantified by an FLA7000 image analyzer. All enzyme assays were performed in triplicate.

### 3.6. Statistical Analysis

Statistical significance was assessed using one-way ANOVA followed by Tukey’s multiple comparison test using GraphPad Prism 8 (GraphPad Software Inc., La Jolla, CA, USA), with *p* < 0.05 considered statistically significant. Data are presented as mean ± standard deviation (S.D.) of the mean.

### 3.7. Western Blotting

The homogenates (30 μg of protein) of the cells expressing FLAG-tagged enzymes were separated by SDS-PAGE on 10% gel and electrotransferred to a hydrophobic polyvinylidene difluoride membrane (Immobilon-P). The membrane was blocked with PBS containing 5% dried skimmed milk and 0.1% Tween 20 (buffer B) and then incubated with anti-FLAG antibody (1:2000 dilution) in buffer B at room temperature for 1 h, followed by incubation with horseradish peroxidase-labeled anti-rabbit IgG antibody (1:4000 dilution) in buffer B at room temperature for 1 h. The membrane was finally treated with Pierce Western Blotting Substrate Plus, and the labeled proteins were visualized with the aid of a LAS1000plus luminoimaging analyzer (FUJIX Ltd., Tokyo, Japan).

### 3.8. Reverse Transcription-PCR

First-strand cDNA was prepared as described in “*4.2. Construction of Expression Vectors*” and subjected to PCR amplification by Quick taq DNA polymerase. The primer sequences used are shown in Table 2. The PCR conditions were as follows: 30 cycles with denaturation at 98 °C for 10 s, annealing and extension at 68 °C for 60 s for cPLA_2_γ; 30 cycles with denaturation at 98 °C for 10 s, annealing and extension at 68 °C for 60 s for ABHD4; 18 cycles with denaturation at 98 °C for 10 s, annealing at 55 °C for 30 s, and extension at 68 °C for 30 s for 18S rRNA as a control.

## 4. Conclusions

In the present study, we first suggested that in living cells, the γ isoform of cPLA_2_ has PLA_1_/A_2_ activity to generate lysoNAPE from NAPE, which was produced by the ɛ isoform of cPLA_2_. We then showed that the purified cPLA_2_γ hydrolyzes not only NAPE to lysoNAPE, but also lysoNAPE to GP-NAE. These consecutive hydrolytic reactions starting from NAPE were previously reported with ABHD4. Considering the wide distribution of cPLA_2_γ in mouse tissues, cPLA_2_γ may function as an alternative of ABHD4, constituting the NAPE-PLD-independent pathway for the biosynthesis of bioactive NAEs.

## Figures and Tables

**Figure 1 molecules-26-05213-f001:**
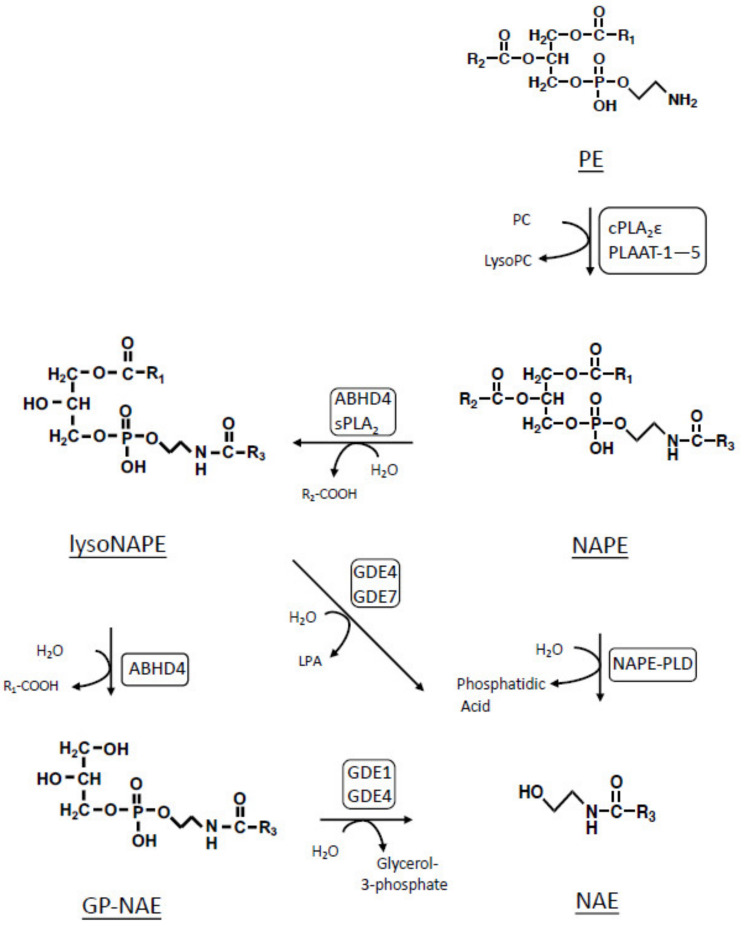
Biosynthetic pathways of NAE in mammals. PLAAT: phospholipase A and acyltransferase.

**Figure 2 molecules-26-05213-f002:**
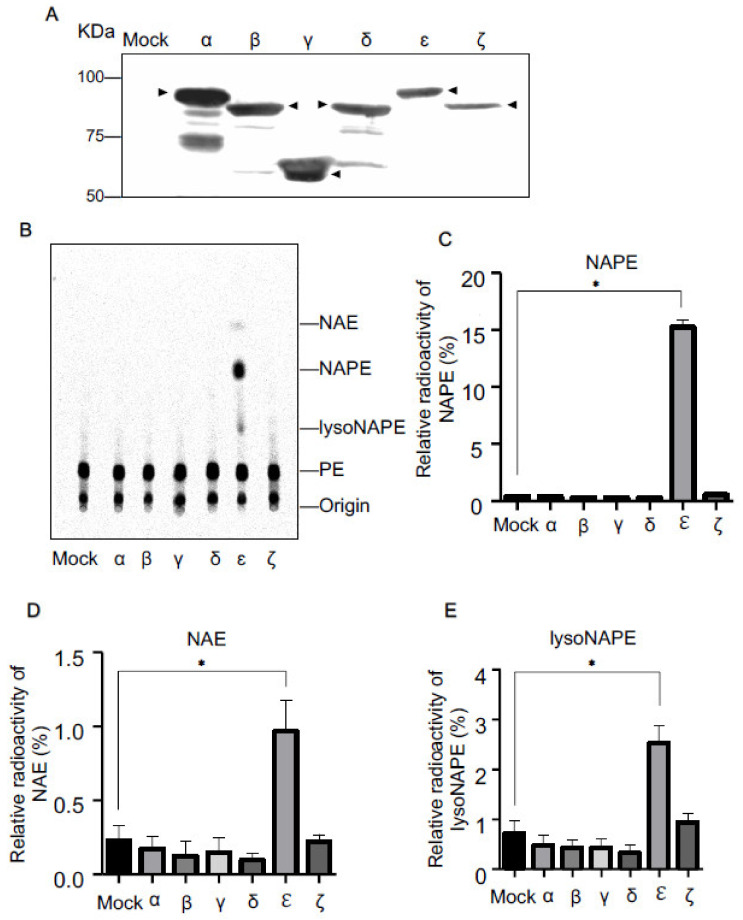
Metabolic labeling of cPLA_2_-expressing cells with [^14^C]ethanolamine. HEK293 cells were transfected with the insert-free vector (Mock) or the expression vector harboring cDNA for the indicated cPLA_2_ isoforms tagged with FLAG. Their expressions were confirmed by Western blotting using anti-FLAG antibody (**A**). Arrowheads indicate the positions of the deduced molecular mass of each isoform. The cells were metabolically labelled with [^14^C]ethanolamine, followed by the treatment with ionomycin. Total lipids were then analyzed by TLC (**B**). The positions of the origin and authentic compounds are shown. The relative radioactivities of NAPE (**C**), NAE (**D**), and lysoNAPE (**E**) are shown (mean values ± S.D., n = 3). * *p* < 0.05.

**Figure 3 molecules-26-05213-f003:**
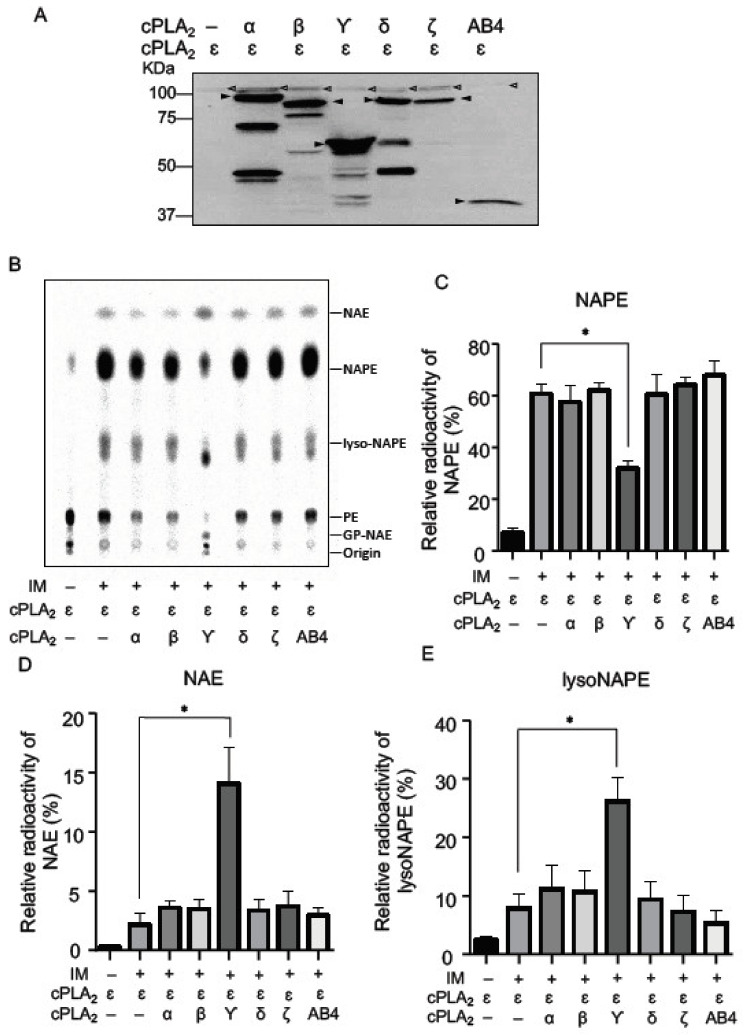
Metabolic labeling with [^14^C]ethanolamine of the cells co-expressing cPLA_2_ε and one of the other isoforms. cPLA_2_ε/Tet-on cells were transfected with the insert-free vector or the expression vector harboring cDNA for the indicated FLAG-tagged cPLA_2_ isoforms or ABHD4 (AB4). Their expressions were confirmed by Western blotting using anti-FLAG antibody (**A**). Arrowheads indicate the positions of the deduced molecular mass of each isoform and ABHD4. The cells were labelled with [^14^C]ethanolamine, followed by the treatment with ionomycin (IM). Total lipids were then analyzed by TLC (**B**). The positions of the origin and authentic compounds are shown. The relative radioactivities of NAPE (**C**), NAE (**D**), and lysoNAPE (**E**) are shown (mean values ± S.D., n = 3). * *p* < 0.05.

**Figure 4 molecules-26-05213-f004:**
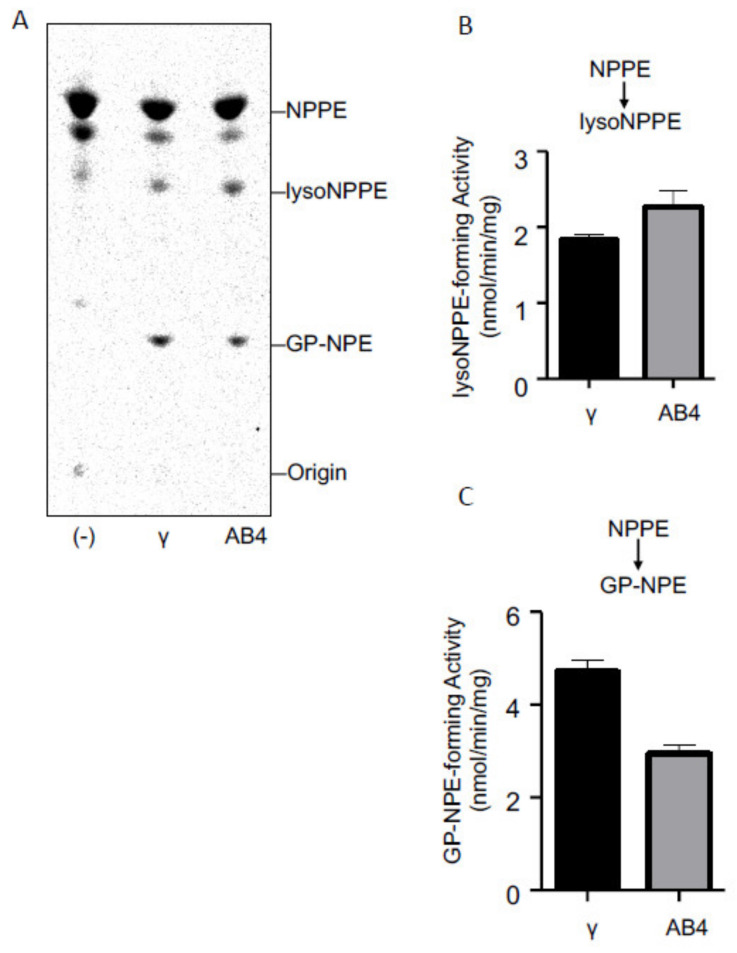
Reactivity of cPLA_2_γ and ABHD4 with NAPE. The purified cPLA_2_γ and ABHD4 (AB4) as well as buffer alone (-) were allowed to react with *N*-[^14^C]palmitoyl-PE (NPPE), and the products were analyzed by TLC (**A**). The positions of the origin and authentic compounds are shown. The *N*-[^14^C]palmitoyl-lysoPE (lysoNPPE)-forming activity (**B**) and GP-*N*-[^14^C]palmitoylethanolamine (GP-NPE)-forming activity (**C**) are shown (mean values ± S.D., n = 3).

**Figure 5 molecules-26-05213-f005:**
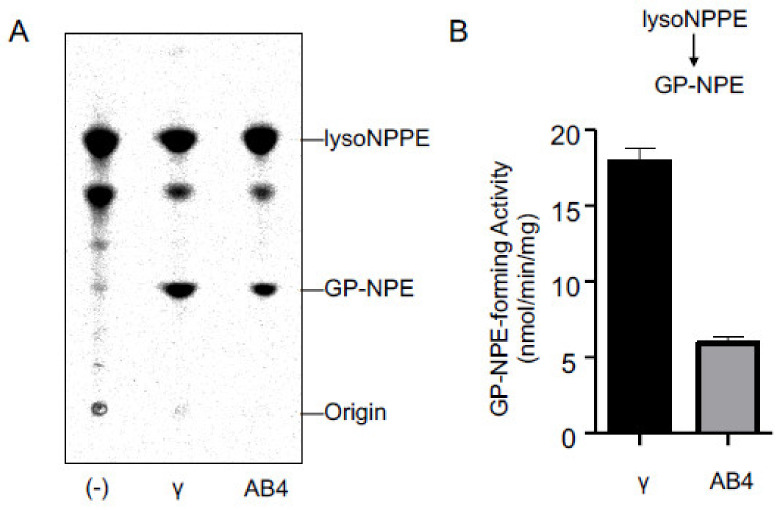
Reactivity of cPLA_2_γ and ABHD4 with lysoNAPE. The purified cPLA_2_γ and ABHD4 (AB4), as well as buffer alone (-) were allowed to react with *N*-[^14^C]palmitoyl-lysoPE (lysoNPPE), and the products were then analyzed by TLC (**A**). The positions of the origin and authentic compounds are shown. The GP-*N*-[^14^C]palmitoylethanolamine (GP-NPE)-forming activity is shown (mean values ± S.D., n = 3) (**B**).

**Figure 6 molecules-26-05213-f006:**
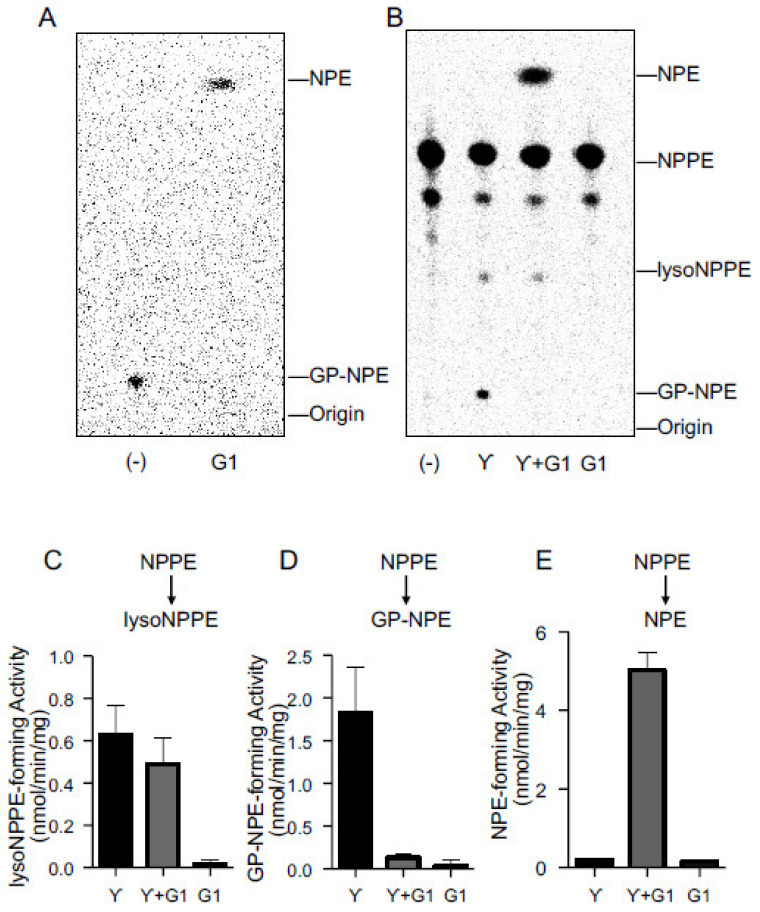
The formation of NAE from GP-NAE by GDE1. The radioactive band corresponding to GP-*N*-[^14^C]palmitoylethanolamine (GP-NPE), which was produced by purified cPLA_2_γ, was scraped from the TLC plate. The radioactive compound was then extracted by the Bligh and Dyer protocol, and incubated with purified GDE1 (G1) or water (-) (**A**). *N-*[^14^C]Palmitoyl-PE (NPPE) was incubated with the purified cPLA_2_γ for 30 min and additionally with the purified GDE1 for 15 min (γ+G1) (**B**). GDE1 or cPLA_2_γ was omitted in γ and G1, respectively. The products were analyzed by TLC. The positions of the origin and authentic compounds are shown. The *N*-[^14^C]palmitoyl-lysoPE (lysoNPPE)-forming activity (**C**), GP-NPE-forming activity (**D**), and *N*-[^14^C]palmitoylethanolamine (NPE)-forming activity (**E**) are shown (mean values ± S.D., n = 3).

**Figure 7 molecules-26-05213-f007:**
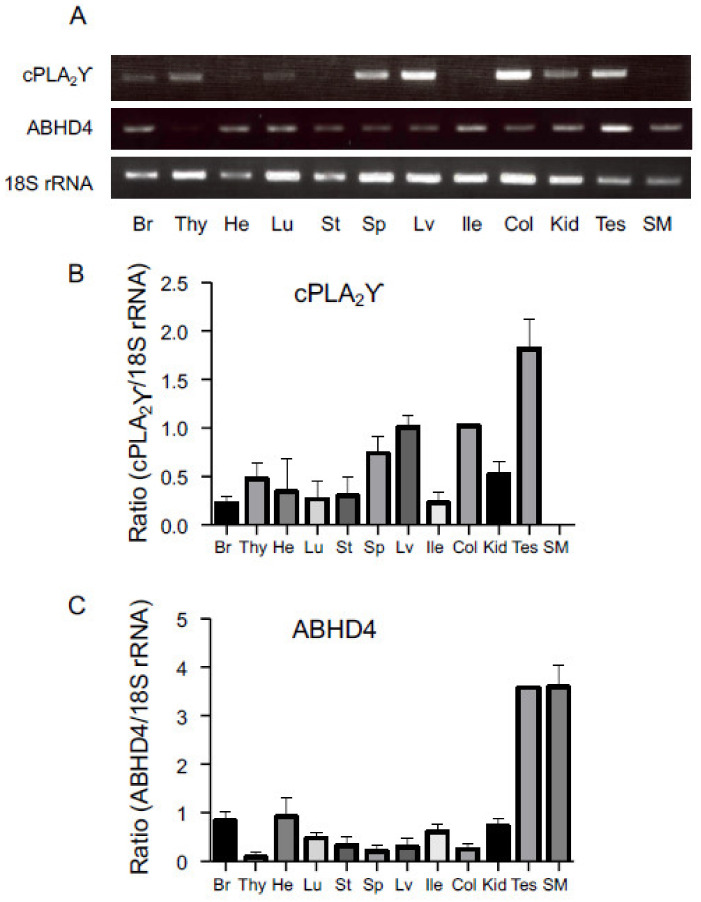
Tissue distribution of cPLA_2_γ and ABHD4 in mice. mRNAs from the indicated mouse tissues were analyzed by reverse transcription-PCR using primers specific for cPLA_2_γ, ABHD4, and 18S rRNA (a control) (**A**). The semi-quantitative results are also shown (mean values ± S.D., n = 3) (**B**,**C**). Br: brain; Thy: thymus; He: heart; Lu: lung; St: stomach; Sp: spleen; Lv: liver; Ile: ileum; Col: colon; Kid: kidney; Tes: testis; SM: skeletal muscle.

**Table 1 molecules-26-05213-t001:** Primers and mouse tissues used for the construction of expression vectors.

cDNA (Accession Number)	Direction	Sequence (Restriction Sites and a Tag Sequence)	Tissue
cPLA_2_α (NM_008869)	forward	5′-cgcactagtggaaaatggattacaaggatgacgacgataagtctttcatagatccttatcagcac-3′ (*Spe* I site, in-frame FLAG sequence)	Thymus
	reverse	5′-cgcgcggccgcttacacagtgggtttacttagaaa-3′ (*Not* I site)	
cPLA_2_β (NM_145378)	forward	5′-cgcactagtggaaaatggattacaaggatgacgacgataaggctctgcaaacctgcccagtctac-3′ (*Spe* I site, in-frame FLAG sequence)	Brain
	reverse	5′-cgcgcggccgctcactccggcctaaactgtttgcg-3′ (*Not* I site)	
cPLA_2_γ (NM_001004762)	forward	5′-cgcactagtggaaaatggattacaaggatgacgacgataaggaactaagctctggggtctgccct-3′ (*Spe* I site, in-frame FLAG sequence)	Brain
	reverse	5′-cgcgcggccgcttaatccttagatatgttgtggga-3′ (*Not* I site)	
cPLA_2_δ (NM_001024137)	forward	5′-cgcactagtggaaaatggattacaaggatgacgacgataagtggagtggagatagaagagtaggc-3′ (*Spe* I site, in-frame FLAG sequence)	Testis
	reverse	5′-cgcgcggccgctcacgtcttcactcccaatggcct-3′ (*Not* I site)	
cPLA_2_ζ (NM_001024145)	forward	5′-cgcactagtggaaaatggattacaaggatgacgacgataagccctggactctccagccaaagtgg-3′ (*Spe* I site, in-frame FLAG sequence)	Large intestine
	reverse	5′-cgcgcggccgctcagcccccaacccttcccccagc-3′ (*Not* I site)	
ABHD4 (NM_134076)	forward	5′-cgcactagtggaaaatggattacaaggatgacgacgataaggctgatgatctggagcagcagcctcag-3′ (*Spe* I site, in-frame FLAG sequence)	Brain
	reverse	5′-cgcgcggccgctcagtcaactgagttgcagatctcttc-3′ (*Not* I site)	

**Table 2 molecules-26-05213-t002:** Primers used for reverse transcription-PCR.

Gene	Direction	Sequence
cPLA_2_γ	forward	5′-tgaggtgagcgaggatcagctgaag-3′
	reverse	5′-atgagtcagatagttttactgtccc-3′
ABHD4	forward	5′-ggcacagtttgggaggattcctggc-3′
	reverse	5′-gaggtgcacggatctcactagggtc-3′
18S rRNA	forward	5′-gtaacccgttgaaccccatt-3′
	reverse	5′-ccatccaatcggtagtagcg-3′

## Data Availability

Not applicable.

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
