# Peer review of "Involvement of the γ Isoform of cPLA_2_ in the Biosynthesis of Bioactive *N-*Acylethanolamines"

_molecules, 2021, doi:10.3390/molecules26175213_

Round 1

Reviewer 1 Report

Point 1. Introduction. The sentence “The results suggested that the γ isoform deacylates NAPE to GP-NAE via the formation of lysoNAPE and constitutes the alternative pathway for the formation of NAE.” (line 64-65) as the only result of the study is in contradiction with the data, presented in the Fig. 2, where the only “cPLA2ɛ-expressing cells exhibited remarkable increases in the intensities of the radioactive bands corresponding to authentic NAPE, NAE, and lysoNAPE” (89-90). Please correct this paragraph to include results with cPLA2ɛ-isoform.

Point 2. Introduction and abstract (15, 36, 38). Please note, that the term “amide’ correspond to the chemical group like –C(O)NH-, whereas “oyl” means –C(O). So, correct names for acylethanolamines will be the follows: Palmitoylethanolamine or Palmitylethanolamide. From the chemical point of view the term “Palmitoylethanolamide” corresponds to –C(O)C(O)NH- structure. Unfortunately, such nuances of chemical nomenclature are often ignored, especially in biochemical articles. Please, correct.

Point 3. Western blot at Fig. 2A needs the line with protein size markers. Please, explain the extra bands in lines α and γ.

Point 4. Western blot at Fig. 3A needs the lines with protein size markers and ɛ-isoform alone. Lines with ɛ-isoform are invisible almost in all cases. Please, provide more contrast picture and discuss in the text the relative expression of members in each pair of proteins. An extra band in the case of α+ ɛ and δ+ ɛ also needs explanation.

Point 5. The authors established the role of the cPLA2 γ-isoform as an additional enzyme in the biosynthesis of acylethanolamines in the mouse. For the same isoform in the human body, only the distribution of γ-isoform in various tissues was mentioned (217). However, the authors do not discuss the question of the similarity of structures and enzymatic activity of mouse and human enzymes. It is necessary to supplement the discussion with the relevant literature data.

Author Response

Reviewer 1

Thank you for your e-mail concerning our manuscript entitled “Involvement of γ isoform of cPLA2 in the biosynthesis of bioactive N-acylethanolamines” (Submission no: molecules-1337687). According to the comments, we carefully revised our manuscript. We will appreciate it very much if you would reconsider this revised version.

Point 1.

Introduction. The sentence “The results suggested that the γ isoform deacylates NAPE to GP-NAE via the formation of lysoNAPE and constitutes the alternative pathway for the formation of NAE.” (line 64-65) as the only result of the study is in contradiction with the data, presented in the Fig. 2, where the only “cPLA2ɛ-expressing cells exhibited remarkable increases in the intensities of the radioactive bands corresponding to authentic NAPE, NAE, and lysoNAPE” (89-90). Please correct this paragraph to include results with cPLA2ɛ-isoform.

Response to Point 1:

According to the comment, we added “as well as the reactivity of purified cPLA2γ with NAPE and lysoNAPE” to the first sentence of the last paragraph in Introduction (line 65 in the revised version).

Point 2.

Introduction and abstract (15, 36, 38). Please note, that the term “amide’ correspond to the chemical group like –C(O)NH-, whereas “oyl” means –C(O). So, correct names for acylethanolamines will be the follows: Palmitoylethanolamine or Palmitylethanolamide. From the chemical point of view the term “Palmitoylethanolamide” corresponds to –C(O)C(O)NH- structure. Unfortunately, such nuances of chemical nomenclature are often ignored, especially in biochemical articles. Please, correct.

Response to Point 2. According to the comment, we changed palmitoylethanolamide (lines 15 and 38), oleoylethanolamide (lines 15 and 39) and arachidonoylethanolamide (lines14 and 36) to palmitylethanolamide, oleylethanolamide and arachidonylethanolamide, respectively.

Point 3.

Western blot at Fig. 2A needs the line with protein size markers. Please, explain the extra bands in lines α and γ.

Response to point 3
In Western blotting shown in Figure 2A, the protein size markers could be visualized as black or white bands by fluorescence (Please see the supplementary figure), while the immunnopositive bands for FLAG peptides were detected by chemiluminescence. Therefore, we did not include the lane of protein markers in Figure 2A. Since the antibody should specifically recognize FLAG peptides, we presume that the extra bands in α and γ lines of Figure 2A are degradative or modified proteins of α and γ isoforms, respectively. We added the sentence “The extra bands were presumed to be degradative or modified proteins of each isoform” to the text (lines 79-80).

Point 4.

Western blot at Fig. 3A needs the lines with protein size markers and ɛ-isoform alone. Lines with ɛ-isoform are invisible almost in all cases. Please, provide more contrast picture and discuss in the text the relative expression of members in each pair of proteins. An extra band in the case of α+ ɛ and δ+ ɛ also needs explanation.

Response to point 4.
We added the line of ε alone to Figure 3A. In Western blotting shown in Figure 3A, the protein size markers could be visualized as black or white bands by fluorescence (Please see the supplementary figure), while the immunnopositive bands for FLAG peptides were detected by chemiluminescence. Therefore, we did not include the lane of protein markers in Figure 3A. We replaced Figure 3A with a more contrast picture (please see the new Figure 3 attached to the end of the manuscript) and discussed the relative expression of isoforms in each pair of proteins in text (lines 115-119). “cPLA2ε cPLA2ε was stably expressed in cPLA2ε/Tet-on cells in the presence of doxycyclin, while other isoforms of cPLA2 were transiently and potently expressed by the introduction of each cDNA using Lipofectamine 2000. The differences in the expression levels between ε and other isoforms were presumably attributed to the differences in the expression methods.” We presume that the extra bands are degradative or modified proteins of each isoform.

Point 5.

The authors established the role of the cPLA2 γ-isoform as an additional enzyme in the biosynthesis of acylethanolamines in the mouse. For the same isoform in the human body, only the distribution of γ-isoform in various tissues was mentioned (217). However, the authors do not discuss the question of the similarity of structures and enzymatic activity of mouse and human enzymes. It is necessary to supplement the discussion with the relevant literature data.

Response to Point 5.

We discussed the similarity of the structure and enzymatic activity of mouse and human cPLA2γ (lines 227-232). “Human cPLA2γ (Accession number, NP_003697) (20) and mouse cPLA2γ (NM_001004762) were deduced to comprise 541 and 597 amino acids, respectively. Arg-54, Ser-82 and Asp-385, forming the catalytic center of human cPLA2γ, was conserved as Arg-55, Ser-83 and Asp-417 in mouse cPLA2γ (17). The human cPLA2γ (20) and mouse cPLA2γ (Guo et al., unpublished observation) showed the PLA1/A2 activity for PC. Thus, human cPLA2γ was considered to be the ortholog of the mouse enzyme.”

Reviewer 2 Report

General comments

The study of Guo and colleagues addresses the involvement of the cPLA2 isoforms other than e in NAE biosynthesis using HEK293 cells. My comments are as follows:

Specific comments:

  • The band corresponding to cPLA2e isoform is barely seen in Figure 3A.
  • Are there specific inhibitors for cPLA2γ and ABHD4?. These pharmacological tools might strengthen the results particularly if used in primary cultures. In the same line, the authors might try to corroborate the tissue distribution of both enzymes at the protein level by western blot.
  • I miss quantitative analysis of results in Figures 6 and 7. How many experiments were performed?
  • A discussion section would be appreciated.

Minor concerns:

The manuscript is well written in general. I would suggest some minor changes:

  • Lines 131: “…despite the fact that purified ABHD6” instead of “…despite that the purified ABHD6”
  • Lines 148: “…showed that both enzymes” instead of “…showed that both the enzymes”

Author Response

Reviewer 2,

Thank you for your e-mail concerning our manuscript entitled “Involvement of γ isoform of cPLA2 in the biosynthesis of bioactive N-acylethanolamines” (Submission no: molecules-1337687). According to the comments, we carefully revised our manuscript. We will appreciate it very much if you would reconsider this revised version.

Point 1.

The band corresponding to cPLA2e isoform is barely seen in Figure 3A.

Response to Point 1.

We replaced Figure 3A with a more contrast picture (please see the new Figure 3 attached to the end of the manuscript). cPLA2ε was stably expressed in cPLA2ε/Tet-on cells in the presence of doxycyclin, while other isoforms of cPLA2 were transiently and potently expressed by the introduction of each cDNA using Lipofectamine 2000. The differences in the expression levels between ε and other isoforms were presumably attributed to the differences in the expression methods.

Point 2.

Are there specific inhibitors for cPLA2γ and ABHD4? These pharmacological tools might strengthen the results particularly if used in primary cultures. In the same line, the authors might try to corroborate the tissue distribution of both enzymes at the protein level by western blot.

Response to Point 2.

Currently, specific inhibitors are not available. We just discussed the usefulness of such inhibitors in distinguishing two enzyme activities in lines 222-224. “Specific inhibitors for these enzymes will be useful to quantitatively estimate the contribution of each enzyme, particularly if used in primary culture.”

Since we don’t have specific antibodies for cPLA2γ and ABHD4, we showed the distribution of their mRNAs in mouse tissues.

Point 3.

I miss quantitative analysis of results in Figures 6 and 7. How many experiments were performed?

Response to Point 3.

We performed the same experiments three times, and showed representative results in the figures. We added this information to the legends of Figures 6 and 7.

The legend of Figures 6: “--- by TLC. The assays were carried out three times and the representative results are shown.”

The legend of Figures 7: “--- (a control). The assays were carried out three times and the representative results are shown.”

Point 4.

A discussion section would be appreciated.

Response to Point 4.

According to the instruction for the journal, we set up “2. Results and discussion” and “3. Conclusion” rather than “Discussion”.

Point 5.

I changed “--- despite that the purified ABHD4 ---” to “--- despite the fact that the purified ABHD4 ---” (line 130).

Response to Point 5.

I changed “--- despite that the purified ABHD4 ---” to “--- despite the fact that the purified ABHD4 ---” (line 136).

Point 6.

Lines 148: “…showed that both enzymes” instead of “…showed that both the enzymes”

Response to Point 6.

I changed “--- both the enzymes ---” to “--- both enzymes --- ---” (line 153).

Round 2

Reviewer 2 Report

I still think that quantificacion is a must in Fig 6 and Fig 7. The authors have addressed the rest of my concerns and modified the manuscript accordingly.

Author Response

Reviewer 2,

Thank you for your e-mail concerning our manuscript entitled “Involvement of γ isoform of cPLA2 in the biosynthesis of bioactive N-acylethanolamines” (Submission no: molecules-1337687). According to the comments, we carefully revised our manuscript. We will appreciate it very much if you would reconsider this revised version.

Point 1

I still think that quantification is a must in Fig 6 and Fig 7.

Response to Point 1:

  • Concerning Figure 6, we quantified the radioactive bands corresponding to lysoNPPE, GP-NPE, and NPE in Figure 6B and showed the results as Figure 6C, D, and E, respectively.

  • Concerning Figure 7, we quantified the bands of cPLA2γ and ABHD4 and showed the results as Figure 7B and C, respectively.
